# Stoichiometry Dependence of Physical and Electrochemical Properties of the SnO_x_ Film Anodes Deposited by Pulse DC Magnetron Sputtering

**DOI:** 10.3390/ma14071803

**Published:** 2021-04-06

**Authors:** Yibo Ma, Xiaofeng Zhang, Weiming Liu, Youxiu Wei, Ziyi Fu, Jiuyong Li, Xuan Zhang, Jingjing Peng, Yue Yan

**Affiliations:** Beijing Engineering Research Center of Advanced Structural Transparencies for the Modern Traffic System, Beijing Institute of Aeronautical Materials, Beijing 100095, China; mayibo555@163.com (Y.M.); xiaofeng.zhang@biam.ac.cn (X.Z.); avic_weiming@126.com (W.L.); qiulinhaizai@163.com (Y.W.); fu19801297313@163.com (Z.F.); 15737936609@163.com (J.L.); kaixuan1226@163.com (X.Z.); pjj.csu.mat@163.com (J.P.)

**Keywords:** tin oxide film, reactive magnetron sputtering, oxygen vacancy, cycle performance

## Abstract

A batch of Sn oxides was fabricated by pulse direct current reactive magnetron sputtering (pDC−RMS) using different Ar/O_2_ flow ratios at 0.3 Pa; the influence of stoichiometry on the physical and electrochemical properties of the films was evaluated by the characterization of scanning electron microscope (SEM), X-ray diffraction (XRD), X-ray reflection (XRR), X-ray photoelectron spectroscopy (XPS) and more. The results were as follows. First, the film surface transitioned from a particle morphology (roughness of 50.0 nm) to a smooth state (roughness of 3.7 nm) when Ar/O_2_ flow ratios changed from 30/0 to 23/7; second, all SnO_x_ films were in an amorphous state, some samples deposited with low O_2_ flow ratios (≤2 sccm) still included metallic Sn grains. Therefore, the stoichiometry of SnO_x_ calculated by XPS spectra increased linearly from SnO_0.0.08_ to SnO_1.71_ as the O_2_ flow ratios increased, and the oxidation degree was further calibrated by the average valence method and SnO_2_ standard material. Finally, the electrochemical performance was confirmed to be improved with the increase in oxidation degree (x) in SnO_x_, and the SnO_1.71_ film deposited with Ar/O_2_ = 23/7 possessed the best cycle performance, reversible capacity of 396.1 mAh/g and a capacity retention ratio of 75.4% after 50 cycles at a constant current density of 44 μA/cm^2^.

## 1. Introduction

Tin oxide (SnO_x_) has been applied as a promising anode since the Fuji Co. first proposed the use of SnO_x_ in lithium–ion batteries [1,2]. Although SnO_x_ possesses better cycle performance than the pure Sn electrode [3], it still suffers the problems of irreversible capacity loss and rapid capacity decay arising from the mechanical failure (e.g., cracking, pulverization and disconnection) during discharge/charge cycles [4,5]. It is known that SnO_x_ would transform into “Li_2_O matrix and metallic Sn” and would be surrounded by solid electrolyte interphase (SEI) during the first lithiation process; Li_2_O could prevent Sn nanoparticles from congesting into large clusters and partially alleviate large volume strain [6]; however, the Li_2_O and SEI layers are not effective electronic conductors and could not store Li^+^ reversibly; both of them would result in increased impedance and irreversible capacity loss [7,8]. In addition, different nanosized SnO_x_ (nanoparticles [9], nanorods [10], nanofibers [11] and nanofilms [12,13,14]) have been devoted to relieving the mechanical failure and providing much shorter Li^+^ transfer paths to improve cycle performance [15]; on the contrary, the increased specific surface area would increase the electrode impedance and exhibit a greater irreversible capacity loss. Therefore, there is a special stoichiometry SnO_x_ corresponding to the best cycle performance.

SnO_x_ nanofilm is one kind of anode used in thin film lithium–ion batteries and can be fabricated by many methods, such as chemical vapor deposition (CVD) [16], radio frequency magnetron sputtering (RF−MS) [14], molecular beam epitaxial (MBE) [17], pulse laser deposition (PLD) [18] and spray pyrolysis (SP) [19,20]. Among these methods, MS has been proven to be an ideal approach for preparing films, since it is simple, manageable and convenient to control the film thickness, composition and structure [21,22]. In addition, reactive magnetron sputtering (RMS, sputtering with introduced reactive gas) could prepare special film composition different from the target, and even control the film composition precisely by adjusting the partial pressure of reactive gas. However, the sputtering mode would lie in a spontaneous conversion between the metal and oxide deposition mode when the oxygen partial pressure increases gradually [23]; then, a transition state in film composition is formed. For instance, the SnO_2_ films prepared by RF−RMS have good cycle performance and show a stable capacity of 440 μAh/cm^2^·μm after 100 cycles at a constant current density of 300 μA/cm^2^ in the range of 0.1–1.0 V [9]. Furthermore, the Ar/O_2_ flow ratio is a key factor determining the oxidation degree of SnO_x_ when conducting the deposition at a fixed working pressure; a set of SnO_x_ films with increasing x values have been fabricated by tuning the Ar/O_2_ flow ratios [11]; it could be inferred that the structural and optical properties dramatically depend on the deviation of composition [20]. However, there is no systematic research illustrating the influence of oxidation degree on electrochemical performance and introducing a gradual evolution when the SnO_x_ composition transforms from a low oxygen ratio to a high oxygen ratio.

In this paper, a group of SnO_x_ films was prepared via pDC−RMS using different Ar/O_2_ flow ratios, and then, all properties, such as morphology, microstructure, surface chemistry and electrochemical properties, were evaluated and compared with each other. Later, a special stoichiometry SnO_x_ was obtained corresponding to the best cycle performance. This work helps us to obtain a better understanding of the O:Sn atomic ratio effect on comprehensive performance.

## 2. Experimental Analysis

The SnO_x_ films were directly deposited on two types of substrates, Cu foils (Dilo Corporation, Dongguan, China, 30 μm thickness, 99.3% purity) and glass sheets (Schoot Corporation, Suzhou, China, sodalime−silicate glass, 1.0 mm thickness). The deposition process was carried out under the following conditions: pure Sn target (Dream Material Technology Corporation, Beijing, China, 99.99% purity, Ф75 × 4 mm) was selected; the target−substrate distance, sputtering pressure, substrate temperature and the sputtering power were fixed at 18 cm, 0.3 Pa, 25 °C and (0.45 W/cm^2^, 100 kHz, 70% of duty cycle), respectively. Then, the SnO_x_ films with different Ar/O_2_ flow ratios (30/0, 29/1, 28/2, 26/4, 25/5, 24/6, 23/7) were fabricated one by one. Other SnO_x_ films with approximate thicknesses of ≈130 μm were prepared later with the aid of deposition rate, as shown in Figure 1b, and these films were then used to characterize transmittance, roughness and other properties.

The physical properties of the deposited films were characterized as follows. First, field emission scanning electron microscopy (FESEM, SU-8010, Hitachi, Tokyo, Japan) and an atomic force microscope (AFM, Dimension Edge, Bruker, Karlsruhe, Germany) were used to measure the film thickness, morphology and roughness; both the surface and cross−section SEM images were taken by operating under vacuum at a voltage of 30 kV, whereas the AFM images were detected in atmosphere with tapping mode. As for the morphology of the cycled electrodes, the half-cells were dismantled in a glove box and were rinsed in acetone to eliminate residual salts, and then, the examination was performed. Second, the crystal structure and film density were analyzed by X-ray diffraction (XRD, D8 Advance, Bruker, Karlsruhe, Germany) and X-ray reflection (XRR) methods, individually; both of them were measured with the same equipment and X-rays sources (Cu Ka, λ = 0.15418 nm, voltage of 40 kV, current of 40 mA), but with different scanning ranges (10°–80° for XRD; 0°–3° for XRR) and analytical methods. Third, X-ray photoelectron spectroscopy (XPS, Quantera SXM, PHI, Chigasaki, Japan) was conducted by using focused monochromatized Al Kα radiation (1486.6 eV) with a beam size of ≈200 μm^2^ at an incident angle of 45°. All the reported binding energy data were calibrated using the position of the carbon contamination (284.8 eV) on the surface of SnO_x_ films. Then, the analysis of XPS data was performed, and the peak deconvolutions were realized by applying the sum of “70% Gaussian−30% Lorentzian” line shapes after Shirley-type background subtraction; the results were also calibrated by the spectra with Ar^+^ etching and the SnO_2_ power standard material (SnO_2_, M = 150.71, CAS:18282−10−5, AR 99.5%, Aladdin Co., Shanghai, China). An additional valence band (VB) spectrum was further measured by XPS using the same Al Kα radiation to reveal oxidation state. Lastly, the optical properties were performed in the wavelength range of 300–800 nm by a UV–Visible spectrometer (Cary 5000, Varian, Palo Alto, CA, USA). 

Standard two-electrode (CR 2025) coin cells were assembled with SnO_x_ film on Cu foil as the working electrode, metallic lithium foil as the counter and reference electrodes and the microporous polyethylene film (PE, 30 μm in thickness, Linyi Gelon New Battery Material Co., Linyi, China) as the separator; hence, the half-cell was set as a configuration of Li metal (−) | separator | SnO_x_ (+). Then, the coin cell was immersed in liquid electrolyte, 1.0 mol/L LiPF_6_ in a mixture of ethylene carbonate (EC), dimethyl carbonate (DMC) and ethyl methyl carbonate (EMC), with the volume ratio of 1:1:1. All coin cells were assembled in a glove box with moisture and oxygen contents of less than 1 ppm. Galvanostatic charge/discharge was carried out using a constant current of 44 μA/cm^2^ between 0.01 and 1.2 V with a battery tester (CT2001A, LAND, Wuhan, China), and the rate performance was performed in the mutative current density range (22–176) μA/cm^2^. CV curves were measured using an electrochemical workstation (650e, CHI, Shanghai, China), with a scanning rate of 0.1 mV/s between 0.01 and 2.0 V; electrochemical impedance spectroscopy (EIS) was then performed with a voltage amplitude of 5 mV in the frequency range of 10^−2^–10^6^ Hz.

## 3. Results and Discussion

### 3.1. Basic Physical Properties of the SnO_x_ Films

It is known that MS is a technique for the preparation of thin film based on gas discharge; the partial pressure of introduced oxygen is critical to the sputtering state when performing RMS from the metallic target [24]. Hence, we first investigated the target voltage as a function of Ar/O_2_ flow ratios at a fixed working pressure, as shown in Figure 1a. The target voltage hysteresis curve of SnO_x_ film was different from that of the SiO_2_ film deposited with the same Ar/O_2_ atmosphere and the pure Si target. The Sn target voltage increased by 30 V abruptly and reached a maximum voltage when O_2_ flow ratios increased from 1.8 to 6.0 sccm; then, it gradually decreased to a voltage of 305 V as the O_2_ flow ratios increased from 6.0 to 7.0 sccm; severe fluctuation did not occur until the O_2_ flow ratio exceeded 7.4 sccm. With the following decrease in O_2_ flow ratios, the target voltage showed a hysteresis in the region between 7 and 0.6 sccm, and finally, the target voltage returned to 288 V; the reason for this is that Ar gas has higher sputtering efficiency than O_2_, and the higher the Ar flow ratio is, the greater sputtering yield is. Therefore, the detected hysteresis curve determined a reasonable sputtering atmosphere (O_2_ flow ratio was less than 7 sccm), which prevented the Sn target from poisoning and maintained a continuous deposition. With the selected Ar/O_2_ flow ratios, the deposition rates shown in Figure 1b were calculated by the film thickness measured from the cross-section images (in Appendix A). It could also be seen that the films deposited quickly when performing with low O_2_ flow ratios (O_2_ ≤ 2 sccm) because of the metal deposition mode, and they gradually attenuated to a stable value of 4.0 nm/min with the increase in O_2_ flow ratios; one turning point (Ar/O_2_ = 26/4), namely, a transition state, laid between the metal and oxide sputtering mode, appeared in the deposition rate curve, where the film composition could be assumed to change obviously.

Figure 1c shows the color of the SnO_x_ films deposited with different Ar/O_2_ flow ratios but possessing similar thicknesses of ≈130 nm; it could be seen that the films deposited with low O_2_ flow ratios (O_2_ ≤ 2 sccm) were opaque because of the high metallic Sn content, whereas the films deposited with high O_2_ flow ratios (≥4 sccm) were transparent for the increased Sn−O ingredient [20]. Moreover, Figure 1d demonstrates the optical transmittance spectra and shows an increased transparency with the increase in O_2_ flow ratios; three curves corresponding to the low O_2_ flow ratios (O_2_ ≤ 2 sccm) were almost overlapped, and the film deposited with O_2_ = 4 sccm especially corresponded to the turning point of the optical properties. A steep absorption edge appeared at a wavelength of about 450 nm for the films deposited with high O_2_ flow ratios. The “blue shift” phenomenon of absorption edge occurred with the increase in oxidation degree, since the band gap of the films deposited with low O_2_ flow ratios was small [25,26], which was caused by the doping energy levels in the Sn oxides [27,28].

XRD patterns of the SnO_x_ films deposited on the glass substrate are shown in Figure 1e, since the signals from the Cu foil substrate disturbed the diffraction peaks largely. Then, the detected peaks were compared with the Powder Diffraction Files (PDF 04−0637 and 04−0673) of the crystal Sn and SnO_2_ [29]. It was confirmed that the films deposited with low O_2_ flow ratios (≤2 sccm) possessed obvious peaks located at 30.6°, 32.2°, 43.9°, 44.9°, 55.3° and 63.8°, all of which were ascribed to the lattice planes of 200, 101, 220, 211, 301 and 400 from metallic Sn, respectively, and its grain size was calculated using Scherrer’s equation; see Figure 1f. In the case of the films deposited with Ar/O_2_ = 26/4, the diffraction peaks completely disappeared and this could be explained by the increased oxidation degree [30]; next, the other three amorphous SnO_x_ films were obtained when the O_2_ flow ratios increased to 5, 6 and 7 sccm. The above XRD patterns demonstrated the disappearance tendency of the Sn crystals with the increase in O_2_ flow ratios; the reasons for the amorphous structures are as follows: (1) the SnO_x_ grain size was too small to be detected when compared with the X-ray coherence length; (2) the weak crystalline signal from SnO_x_, such as (110), was covered by the substrate peak positioned at 15°–32° [31]; (3) and the most likely reason is that the deposited materials had low reactivity and could not be arranged in an orderly manner when the substrate was not artificially heated [32]. It is worth noting that the amorphous structure could cause the volumetric strain introduced by alloying/dealloying reactions to be distributed evenly and avoid the local strain in a specific direction for crystal materials.

The SEM images of the as-grown SnO_x_ films are shown the in left part of Figure 2. The SnO_x_ films deposited with low O_2_ flow ratios (O_2_ ≤ 2 sccm) showed particle morphology; for instance, the film deposited with Ar/O_2_ = 30/0 possessed irregularly shaped particles, and its minimum and maximum particle sizes were equal to 100 and 400 nm, respectively. Then, the particles transformed into sphere-like shapes, and the average particle size decreased from 80 to 30 nm when O_2_ flow ratios increased from 1 to 2 sccm; these particles are proved to be composed of metallic Sn by the EDS information in Appendix A. However, the above particle morphology changed into a smooth surface when the O_2_ flow ratio exceeded 4 sccm, as shown in Figure 2d–g; a smooth and dense accumulation film occurred, and no obvious particle was directly observed under the same SEM magnification. The reason for these obvious metallic Sn particles in the films deposited with low O_2_ flow ratios (O_2_ ≤ 2 sccm) is that a large number of Sn particles were sputtered on the substrate, and the particles had strong mobility to form large clusters. However, for the films deposited with relatively high O_2_ flow ratios, the sputtered Sn particles were quickly oxidized, and the Sn−O bonds were formed; therefore, these films exhibited the characteristics of typical amorphous oxide material. Furthermore, the particles measured by SEM (Figure 2h) had a larger average size than that derived from the XRD in Figure 1f, which is attributed to the significant agglomeration of the particles (secondary particles) [33,34].

For the films after performing cycling, many cracks appeared on SnO_x_ films after 20 cycles, as shown in the right part of Figure 2. In Figure 2a’, the film deposited with Ar/O_2_ = 30/0 retained a small amount of metal Sn. In Figure 2b’ and c’, the films deposited with low O_2_ flow ratios (O_2_ = 1, 2 sccm) broke into irregular shapes; obvious cracks (greater than 200 nm in width) and separated “islands” (the size is close to 1 μm) occurred. As for the films deposited with high O_2_ flow ratios (O_2_ ≥ 4 sccm) in Figure 2d’–g’), two kinds of cracks occurred, one type of crack had a longer size and broke up the whole film into many “islands”, and the crack length would become shorter as the O_2_ flow ratios increased; the other type of crack had a smaller size (30 nm in width) and separated the residual part into smaller particles, and these particle size decreased from 46 to 20 nm as the O_2_ flow ratios increased. All cracks were caused by the tensile stress introduced by the conversion reaction (SnO_x_ + Li^+^ + e^−^ → Sn + Li_2_O) during the initial lithiation process, because metallic Sn exhibited a volume of 27 Å^3^, while SnO and SnO_2_ had volumes of 36 Å^3^ and 35 Å^3^, respectively [35]; it could be concluded that metallic Sn occupies 35% less volume compared to SnO_x_, and the reduced volume is the ultimate origin of tensile stress. Meanwhile, the element rearrangement could also be characterized using EDS information (Appendix A); it was found that the residual “islands” were mainly composed of Sn elements, and their distribution was consistent with the “islands” contour; there were no other kinds of elements detected in the crack positions. The Sn element rearrangement was likely a consequence of the high mobility [28] of the nanoscale Sn particles formed in the conversion reaction and the high mobility of Li^+^ in the Li_x_Sn alloy (diffusion constants between 10^−7^ and 10^−8^ cm^2^/s [36]). Recently, researchers have introduced work function (WF) as a parameter to study the stability of electrodes [37,38,39,40,41]. The WF (Φ) is usually defined as the energy required to take away one electron from the Fermi level (*μ*), namely, Φ *= φ − μ*, where *φ* refers to the vacuum level. In the case of VO_x_, the WF increases simultaneously with the oxidation state ranging from the value of 4.3 eV of the pure vanadium up to ≈7 eV as in stoichiometric V_2_O_5_ [38]. Therefore, WF could also be used as a monitoring parameter to determine the stabilization of electrodes. For instance, metallic lithium could be implemented as an anode in a Li-based battery. The observed decrease in the WF indicates an increase in the Fermi energy relative to the vacuum level, with an inevitably enhanced energy drive for the treated surface to lose electrons and reduce interacting species. For another anode, LiTiO_2_, the O vacancies on the surface decrease the WF and promote the interface reaction, the presence of O vacancies on the surface increases the electrons on the surface Ti ions, resulting in the presence of Ti^3+^ or other Ti ions with even lower valences. As a result, the WF decreases, easing the loss of electrons on the surface [39]. In conclusion, WF is an important parameter in Li-based batteries to improve electrode/electrolyte interface reactions.

In order to determine the deposited film quality and calculate the mass specific capacity later, X-ray reflectance (XRR) was adopted to measure the film density. First, Figure 3a shows the relationship between the surface roughness and the Ar/O_2_ flow ratios. This demonstrates that the film deposited with Ar/O_2_ = 30/0 possessed the highest roughness of 50.0 ± 7.0 nm, and the roughness sharply decreased to 4.6 ± 0.2 nm and stabilized at 3.7 ± 0.1 nm with the increase in O_2_ flow ratios. Second, Figure 3b–g show the fitting results of the periodic oscillation curve; black dots represent the original experimental points, and the red curve signifies the fitting curve, exhibiting excellentconsistency with the original data. The films deposited with the condition of O_2_ ≥ 2 sccm possessed roughness of less than 4.5 nm; their reflectance curves shown in Figure 3c–g had periodic oscillation characteristics and could be fitted, the fitting thickness was close to the actual thickness of 130 nm and its relative deviation was less than 8%; the fitting roughness also distributed in the interval of 3.07–4.22 nm, consistent with the roughness results characterized by AFM in Figure 3a and Appendix A. Lastly, the fitting density decreased from 6.58 to 6.20 g/cm^3^ when the O_2_ flow ratios increased from 2 to 7 sccm. The film density was proportional to the critical angle θc in the reflectance curve, and θc showed a decreasing trend as the O_2_ flow ratios increased, confirming the decreased trend of film density displayed in Figure 3h. The film density results fitted by the model shown in Appendix A was close to that of the SnO_2_ films prepared by the atomic layer deposition (ALD) method [28], but still smaller than the bulk densities of SnO of 6.45 g/cm^3^ [42] and SnO_2_ of 6.38 g/cm^3^ [43]. Third, the films deposited with low O_2_ flow ratios had roughness of larger than 4.5 nm, which would cause X-ray scattering and disturb periodic oscillation signals; therefore, the film density could not be directly fitted by the XRR model in Appendix A. The weighing method was then applied, and the densities were calculated to be 7.20 and 6.69 g/cm^3^, respectively. Finally, the values of film density are summarized in Figure 3h; they were always less than the corresponding density of bulk material. For example, the density of the Sn film deposited at Ar/O_2_ = 30/0 was smaller than 7.29 g/cm^3^ of the bulk metallic Sn^3^, which is because the films deposited with the MS method could not reach 100% close packing state; there were many gaps within the as-grown films, as shown in Figure 2. The deposited films belonged to Sn oxides with oxygen vacancy, and the actual densities of the films deposited at various O_2_ flow ratios were lower than the theoretical density.

### 3.2. Composition of the SnO_x_ Films

The types of elements contained in the SnO_x_ films were determined to be C, Sn and O according to the survey spectrum shown in Appendix A. The Sn 3d spectrum is presented in Figure 4a; the Sn 3d core level peak was composed of double peaks, corresponding to Sn 3d3/2 and Sn 3d5/2 at ≈495.0 and ≈486.6 eV, respectively. Furthermore, the spin–orbit splitting between the two peaks was close to 8.4 eV, consistent with the energy levels reported for SnO_2_ [44]. Additionally, it was noted that the SnO_x_ films deposited with low O_2_ flow ratios (O_2_ ≤ 2 sccm) exhibited an obvious shoulder peak at a binding energy close to 485.0 eV, which corresponded to the Sn−Sn bond from metallic Sn^0^ [6,14], and the peak area of Sn^0^ gradually decreased with the increase in O_2_ flow ratios. The existence of Sn^2+^ and Sn^4+^ components was further confirmed by the fitting procedure of the Sn 3d5/2 peak. The Sn 3d5/2 peak can be divided into two peaks corresponding to Sn^2+^ (binding energy of 487.2 eV) and Sn^4+^ (486.4 eV) [15,45,46,47], indicating the coexistence of the SnO and SnO_2_ phases; the energy separation of ≈0.8 eV between Sn^2+^ and Sn^4+^ is interpreted as a chemical shift [44], located in the range of 0.5–0.8 eV, pointed out by the literature [6,48]. It was also noted that the peak position of Sn 3d shifted to a lower binding energy, and the peak shape became gradually symmetrical when the O_2_ flow ratios exceeded 4 sccm. Additionally, the film composition of the SnO_x_ film deposited with Ar/O_2_ = 26/4 was in a transition state, consisting of a mixture of Sn, SnO, SnO_2_.

As shown in Figure 4b, a decomposition procedure of O 1s core level peak was performed to analyze the oxidation degree; it was observed that the O 1s peak in the SnO_x_ films deposited with low O_2_ flow ratios (O_2_ ≤ 2 sccm) was wide, asymmetrical and had a significant shoulder at the higher binding energy (≈532.0 eV) side, while the peak shape gradually became symmetrical when the O_2_ flow ratios increased. According to the peak fitting results, the following conclusions were obtained: (1) The oxygen elements at the shoulder positionbelong to the chemisorbed oxygen (O_chem_), such as hydroxyl ions (OH^−^) [42,45,49,50], and the presence of O_chem_ is not surprising when considering the film morphology with microgaps (like in Figure 2), which can allow gaseous contaminants to attach onto internal particles within the film. Moreover, two oxidation states corresponding to the Sn^4+^ (SnO_2_) and Sn^2+^ (SnO) are easily distinguished. (2) The oxygen element at the dominant peak of ≈530.9 eV comes from lattice oxygen (O_L_), namely, the Sn−O−Sn bond [42,44]. (3) Another one resolved at ≈530.1 eV is associated with the O^2−^ ions in the oxygen-deficient regions (O_V_) [48,51]; O_L_ and O_V_ are also separated by ≈0.8 eV, and this is attributed to the incomplete oxidation state [45]. This result suggests that the oxidation of the Sn component is incomplete under the selected Ar/O_2_ atmosphere, leading to the coexistence of SnO and SnO_2_ phases within the deposited films.

In fact, it was found that the peak positions in the films deposited with different Ar/O_2_ flow ratios were not completely consistent. For example, the binding energies corresponding to Sn^2+^ shifted, and the deviation among the different decomposition results was less than 0.2 V; this binding energy shift could be explained by the buried electrical potential model [52,53], which assumes that the deviation is derived from inhomogeneous charge distribution brought by the mixture composition (Sn, SnO and SnO_2_), the conductive mental Sn, nonconductive SnO_2_ materials, as well as different surface roughness and porosity. All components together led to uneven charge distribution on the film surface, resulting in irregular peak shifts. Moreover, the SnO_x_ films would inevitably come into contact with the air and form surface contamination before XPS testing, since the existing SnO_2_ component is an n-type semiconductor, and the oxygen chemisorptions have a stronger affinity for electrons, so the electrons on the surface were captured and formed a space charge layer [54], which also caused the uneven distribution of the charges. Hence, we performed an Ar^+^ etching experiment [55] and the XPS spectra of the SnO_2_ standard material to calibrate the O:Sn atomic ratios, as shown in Appendix A.

An additional valence band (VB) was further selected to identify the oxidation degree. For instance, the VB spectrum of the sample deposited at Ar/O_2_ = 23/7 is displayed in Figure 4e, and four peaks occurred: (1) the first peak at the binding energy of ≈5.0 eV (I) is ascribed to the O 2p bonding or nonbonding characteristic; (2) intermediate states around 7.5 eV (II) originate mainly from hybridization between Sn 5p and O 2p orbital; (3) the lower part of VB around 10.6 eV (III) is due to the strong interaction between Sn 5s and O 2p orbital, and above three peaks are fingerprints of SnO_x_ [56]; (4) a small band gap state between 2.0 and 4.0 eV (IV) with respect to the Fermi edge was also observed, which is attributed to the band gap states in the SnO_2_ [57,58,59]. Therefore, we can conclude from this VB measurement that the above SnO_x_ films deposited by different Ar/O_2_ atmospheres contain a mixture phase of Sn^4+^ and Sn^2+^.

In order to quantify the oxidation degree (x) in the SnO_x_ film, we used the peak fitting results and factor analysis method (namely, formula 1.1) to calculate the relative content and average valence. The related literature [60,61] have introduced calculating procedures by the atomic sensitivity factor (ASF); for instance, Kwoha et al. [62] calculate the relative O:Sn concentration in SnO_2_ films based on the ASF. In Figure 4c and d, the relative content of Sn^0^ decreased linearly from 16% to 0%, and the content of Sn oxides (Sn^2+^, Sn^4+^) increased when the Ar/O_2_ flow ratios changed from 29/1 to 23/7; the higher the O_2_ partial pressure is, the greater the formation of O_L_ and O_V_ is; the relative content increased from 70% to a stable value of 80% as the O_2_ flow ratios increased. Next, the O:Sn atomic ratios as a function of Ar/O_2_ flow ratios is summarized in Figure 4f; the values of the O:Sn ratio increased by ≈0.19 when the O_2_ flow rate increased by 1 sccm. The SnO_x_ films deposited with low O_2_ flow ratios (O_2_ ≤ 2 sccm) possessed a composition of SnO_0.70_ and SnO_0.74_, respectively, and the O:Sn value was less than 1.0 due to the presence of metal Sn, consistent with the above XRD and SEM results. Specially, the detected oxygen component in the metallic Sn film deposited with Ar/O_2_ = 30/0 likely resulted from the surface oxidation when it transferred through air for the XPS measurement [63]. While other films deposited with high O_2_ flow ratios (O_2_ ≥ 5 sccm) retained the film composition of SnO_1.30_, SnO_1.48_ and SnO_1.71_, the O:Sn value was greater than 1.0 due to the increased oxidation state. The formation of SnO_1.09_ was measured for the film deposited at Ar/O_2_ = 26/4, belonging to a special transition state. However, a thoroughly oxidized SnO_2_ film could not be obtained under the selected Ar/O_2_ atmosphere conditions; this is restricted by the characteristics of the MS process. The introduced reactive O_2_ as an oxygen source was easily pumped away in the sputtering chamber, which reduced the contact with the sputtered Sn particles; meanwhile, the films deposited by the Ar/O_2_ mixture gas tended to be in an oxygen-deficient state owing to insufficient decomposition of O_2_ [20].
(1)$$x=\frac{n_{O}}{n_{Sn}}=\frac{I_{O}/ASF_{O}}{I_{Sn}/ASF_{Sn}}$$
where the stoichiometry *x* is equal to *n_O_/n_Sn_*, representing the atomic ratio; the letter *I* denotes the intensity of the photoelectron (namely, refers to the peak area of a characteristic peak), which has a linear relationship with the atomic concentration in the sample; and the ASF values for Sn and O were confirmed to be 4.095 and 0.711, respectively, according to the library of ASF in the XPS apparatus.

### 3.3. Electrochemical Performance of the SnO_x_ Films

The choice of potential window is crucial for Sn-based compounds to realize high performance, and the influence of different O:Sn atomic ratios on its electrochemical reaction process is still not clear; hence, CV was conducted to study the reaction mechanism for SnO_x_/electrolyte/Li configuration, as presented in Figure 5. All CV curves were measured at a scan rate of 0.1 mV/s between 0.01 and 2.0 V. In order to eliminate the interference from the Cu foil substrate during the electrochemical reaction, Figure 5a first shows the CV curve of bare Cu foil with the same characterized condition, and no reduction/oxidation peaks occurred, which indicates that the Cu foil remained electrochemically inert and did not participate in any reaction under this measurement condition. Then, the other CV curves presented in Figure 5 revealed the electrochemical reaction trend between Li and SnO_x_ when the SnO_x_ evolved as follows: “metallic Sn → oxide deficient → oxide rich”. 

First, the CV curves of SnO_x_ films deposited with low O_2_ flow ratios (O_2_ ≤ 2 sccm) were similar to that of the metallic Sn, exhibiting the independent and sharp peak characteristics just like in Figure 5b–d. Four reduction peaks appeared at 1.65, 0.67, 0.55 and 0.40 V during the first reduction (lithiation) process; the peak at 1.65 V disappeared in the following cycles, and it is ascribed to the decomposition of the electrolyte (the electrolyte is reduced into impurities, such as Li_2_CO_3_ [42]) and the formation of the SEI layer; the reduction peaks at the other three positions corresponded to the Li−Sn alloying reactions, it would form Li−Sn alloys with a high lithium content. While in the following oxidation (delithiation) process, four oxidation peaks appeared at 0.45, 0.60, 0.71 and 0.79 V, respectively, the Li_x_Sn alloy could be in a complete delithiation state when the potential increased to ≈1.0 V. As for the second cycle process, two reduction peaks at 0.67 and 0.55 V merged into one reduction peak at 0.62 V, indicating that the electrode structure changed after the first cycle, but the four oxidation peaks hardly changed and corresponded to the delithiation process of Li_22_Sn_5_, Li_7_Sn_2_, Li_7_Sn_3_ and LiSn [64].

Second, the SnO_x_ films deposited with high O_2_ flow ratios (O_2_ ≥ 4 sccm) containedelectrochemical conversion reaction due to the obvious existence of oxides. The CV curves of the initial three cycles are shown in Figure 5e–h. Taking the film deposited with the condition of Ar/O_2_ = 23/7 as an example, during the first cathodic sweep, the CV traces showed the structural destruction of SnO_x_ above 1.0 V vs. Li/Li^+^ [65]; the broad peaks at 1.23 and 0.96 V represented the decomposition of the electrolyte to form the SEI layer and the conversion reaction of SnO_x_, namely, the formation of intermediate Li_x_SnO_2_ phase [6] first, and then reduced into metallic Sn^0^ particles as shown in formula 1.2 and 1.3 [3,66]. Three other reduction peaks at 0.60, 0.35 and 0.26 V corresponded to the Li^+^ insertion into the Sn structural lattice to form Li_x_Sn alloys, such as Li_0.4_Sn, LiSn and Li_2.33_Sn, according to formula 1.4 [3]. In the first anodic sweep, oxidation peaks at 0.45, 0.62 and 0.75 V corresponded to the delithiation reactions of Li_x_Sn alloys at different stages; these peaks agree well with those given in the metallic Sn [29], and two additional broad peaks at 1.24 and 1.77 V occurred in the oxygen-rich SnO_x_ films. These two newly appeared oxidation peaks corresponded to the partially reversible delithiation reaction of Li_2_O [42]. In subsequent cycles, two cathodic peaks at 1.23 and 0.96 V disappeared, and the other three cathodic peaks were replaced with less intense ones along with the peak position shifts to the low potential direction, which was caused by the increase in battery internal resistance. The battery internal resistance came from two aspects: (1) the formation of Li_2_O during the first lithiation, which is not a good electronic conductor [29], (2) many cracks appeared after cycling; the electrochemically active materials around the crack positions lost electrical contact and formed new SEI layers; the thick SEI layer predominantly contained inorganic products, such as LiF and Li_2_CO_3_, and led to a large resistivity [67]. While in the following oxidation process, the intensity of the three peaks remained unchanged, except for the intensity of the oxidation peak positioned at 0.62 V. The CV curves after the first cycle were completely consistent with each other, indicating a good reversibility for the SnO_x_ film electrodes. 

Insertion (intermediate phase):(2)$$SnO_{2}+xLi^{+}+xe^{-}\to Li_{x}SnO_{2}$$

Conversion: (>1.2 V vs. Li/Li^+^, with capacity of ≈711 mAh/g):(3)$$SnO_{2}+4Li^{+}+4e^{-}\to 2Li_{2}O+Sn$$

Alloying/dealloying: (<0.5 V vs. Li/Li^+^, with capacity of ≈783 mAh/g):(4)$$Sn+xLi^{+}+xe^{-}↔Li_{x}Sn(0\leq x\leq 4.4)$$

Comparing the above CV curves shown in Figure 5, we could notice that the changes in electrochemical reaction revealed a transition from “single alloying” mode to “conversion + alloying” mode owing to the composition transition from “oxide deficient” to “oxide rich” [6]. The core difference in the electrochemical reaction between Sn and SnO_x_ is also given: (1) SnO_x_ films possessed a characteristic reduction peak in the voltage range of 1.3–0.9 V, and the intensity of this peak increased with oxidation degree. This position corresponded to the formation of the SEI layer in the SnO_x_ material and was very different from that in the Sn material. The metallic Sn provided a high voltage of 1.6–2.5 V for SEI formation, which has been testified by an in situ AFM observation [68]. (2) The reduction peaks located in the range of 0.80–0.01 V corresponded to a series of Li−Sn alloying reactions, and the peak shape changed abnormally with the increase in the cycle numbers. For example, the SnO_x_ film deposited at the condition of Ar/O_2_ = 26/4 (Figure 5g) showed a special peak at 0.21 V, and its intensity increased with the cycle number particularly; this is explained by the cluster growth of metallic Sn nanoparticles. The formed Sn nanoparticles after the first cycle agglomerated during the following charge/discharge process due to its good mobility. (3) New oxidation peaks appeared above 1.0 V, which are ascribed to the statement that “Li^+^ are partly released by Li_2_O”; this view has been proved by Mössbauer spectroscopy [42,69] and could be used to explain why the first discharge capacity of SnO_x_ was higher than the theoretical capacity.

Lastly, it was noted that the CV curves measured by the SnO_x_ electrodes were different from those from SnO_x_ powder materials in the literature. More peak signals could be detected in the film electrode, and this film anode is more likely to be an ideal model for analyzing reaction mechanisms. The reaction mechanism of SnO_x_ was confirmed as follows. The electrolyte solution and Li^+^ first reacted at the interface of the SnO_x_/electrolyte to form SEI layers; then, the Li^+^ passed through the SEI layer and penetrated into the SnO_x_ to form intermediate Li_x_SnO_2_. Li_x_SnO_2_ conversed into metallic Sn nanoparticles and the Li_2_O matrix quickly. Next, Li^+^ and the newly formed metallic Sn nanoparticles carried out a series of alloying reactions to form the Li_x_Sn alloy along with Li_2_O distributed around them. During the dealloying process, the Li_x_Sn alloy was completely delithiated at a voltage below 1.0 V, and even part of Li_2_O was reversibly delithiated when the delithiation voltage was higher than 1.0 V. 

The initial discharge/charge curves of all SnO_x_ film electrodes are compared in Appendix A. The curves exhibited the following two characteristics when O_2_ flow ratios increased: (1) The SnO_x_ films deposited with low O_2_ flow ratios (O_2_ ≤ 2 sccm) had a series of obvious potential platforms below 1.0 V, coinciding well with the oxidation/reduction peaks in the above CV curves. However, the SnO_x_ films deposited with high O_2_ flow ratios (O_2_ ≥ 4 sccm) only showed sloping curves without obvious potential platforms. (2) New platforms located at ≈1.2 V were only detected in the SnO_x_ films deposited with high O_2_ flow ratios (O_2_ ≥ 4 sccm), which indicates a conversion reaction. In addition, all SnO_x_ electrodes showed drastic irreversible capacity (≈50%), such as the SnO_1.68_ film deposited at Ar/O_2_ = 23/7, the initial discharge/charge capacity of which was equal to 1035.2/507.1 mAh/g, corresponding to a small coulomb efficiency of 49.0%. Generally, huge irreversible capacity loss in SnO_x_ is attributed to the irreversible Li^+^ loss from the formation of SEI and Li_2_O [28]. The phenomenon that the detected discharge capacity (>1000 mAh/g) is larger than the theoretical capacity (782.0 mAh/g) of SnO_2_ could be explained by the selected cut-off potential of 1.2 V exceeding 0.8 V vs. Li/Li^+^, which causes some Sn^0^ nanoparticles to not be oxidized into SnO_2_, namely, the maximum binding of 6.4 mol of Li^+^ (theoretical capacity of ≈1138 mAh/g) [62].

Then, we compared the cycle performance of all SnO_x_ electrodes. The galvanostatic discharge/charge process was executed in a voltage range of 0.01–1.2 V at a current rate of 44 μA/cm^2^, and the results are shown in Figure 6. The left column represents the specific discharge/charge curves of the 1st, 2nd, 5th, 25th and 50th cycles; the right column represents the discharge capacity and coulomb efficiency as a function of cycle numbers. For the SnO_x_ films deposited with low O_2_ flow ratios (O_2_ = 0, 1, 2 sccm), the reversible capacity after 50 cycles could be delivered as 27.4, 47.3 and 166.4 mAh/g, respectively, while for the SnO_x_ films deposited with increased O_2_ flow ratios (O_2_ ≥ 4 sccm), the relative stable capacity of 379.6, 359.6, 388.6 and 396.1 mAh/g could be delivered after 50 cycles; the maximum reversible capacity of 396.1 mAh/g and capacity retention ratio of 75.4% pertain to the SnO_1.71_ film deposited with Ar/O_2_ = 23/7. The excellent cycle stability in the oxygen-rich film may be attributed to the stable structure and morphology introduced by the elevated oxidation degree. Its amorphous structure (Figure 1e) caused the film volume to expand in all directions uniformly, instead of an individual expansion in the crystal; the smooth surface (Figure 2) was conducive to uniform charge distribution and prevented the local expansion during the alloying/dealloying reactions; and the enhanced oxidation degree in SnO_x_ formed more Li_2_O matrix during cycling. The evenly distributed Li_2_O around Sn nanoparticles plays a very important role in allowing the electrode to expand and contract reversibly. 

However, all SnO_x_ film electrodes possessed a common phenomenon, namely, the discharge capacity decayed with the increase in cycle number. The comprehensive effects of volume change and continuous SEI formation contributed to this capacity decay. The initial SEI layer was formed instantaneously when electrodes contact with the electrolyte solution [70], then the formed Li_x_Sn alloys showed a severe volume expansion effect (up to 300%) and destroyed the entire membrane structure, so the fresh active material was continuously exposed to the electrolyte solution, resulting in further electrolyte decomposition and thick SEI layers [71,72]. Therefore, the reversible capacity did not stop decreasing until the film electrode formed a stable structure, covered by a thick SEI layer or decomposed into a critical size, which would prevent the continuous exposure of the fresh electrode to the electrolyte solution. To confirm our assumption about the relationship between capacity fading and volume change, the morphological evolutions of the SnO_x_ film after 20 cycles were also characterized in Figure 2b; the cracks were generated due to the huge volume variation of “Li_4.4_Sn (7738.9 Å^3^) ↔ Sn (108.2 Å^3^) ↔ SnO (69.8 Å^3^) ↔ SnO_2_ (71.5 Å^3^)” [73,74] during the charge/discharge cycle; the destroyed electrode structure verified that the gradual decay of discharge capacity was related to the detachment of the active materials from the current collector [29]. 

Figure 7 showed the rate capabilities of the SnO_x_ films deposited with different Ar/O_2_ flow ratios, and these electrodes were continuously cycled without suspension among the progressive current densities (44 → 176 → 22 μA/cm^2^). The rate capability of the SnO_x_ films deposited with high O_2_ flow ratios (O_2_ ≥ 4 sccm) was better than that of the films deposited with low O_2_ flow ratios (O_2_ ≤ 2 sccm). For instance, the film deposited with Ar/O_2_ = 23/7 performed cycling at four increased current densities, 44, 88, 132, 176 μA/cm^2^, and carried out 10 cycles at each current density; the final discharge capacity was 412.0 mAh/g, and the capacity retention rate was 79.0%. However, the other electrode had a reversible discharge capacity of 414.1 mAh/g and a capacity retention rate of 79.5% when cycling at a constant current density of 44 μA/cm^2^ after 40 cycles, which indicates that the increased current density had no obvious effect on the reversible discharge capacity. However, the film deposited with Ar/O_2_ = 28/2 only had a discharge capacity of 240.7 mAh/g and a capacity retention rate of 58.7% after 40 cycles. Moreover, the specific capacity did not recover as expected when the current density began to decrease at the 41st cycle. Taking the film electrode deposited with a condition of Ar/O_2_ = 23/7 as an example, it had a reversible discharge capacity of 405.4 mAh/g and a capacity retention rate of 77.8% after the following 30 cycles at the decreased current densities of 132, 88 and 44 μA/cm^2^. The reason for the improvement in the rate capability of the SnO_x_ films deposited with high O_2_ flow ratios is similar to that explaining the improved cycle performance: both could be attributed to the increased Sn^4+^ composition, which plays an important role in electrochemical performance improvement.

The impedance of all SnO_x_ electrodes at different cycling states was also shown in Figure 8. The EIS curves were measured before cycling, after 1 cycle and 20 cycles at 1.2 V vs. Li/Li^+^. All curves were fitted with the same equivalent circuit models shown in Figure 8h. It was found that the Nyquist plots typically displayed a depressed semicircle at the high frequency range, while the sloping line was at the low frequency range. In the equivalent circuit model, Rs is the ohmic resistance of the electrolyte, separator and electrode; R1 and C1 represent the resistance and capacitance of the SEI layer, respectively, corresponding to the semicircular arc in the high frequency region; R2 and C2 represent the double layer capacitance and passivation film capacitance, respectively; W_d_ represents the Warburg impedance caused by Li^+^ diffusion in the electrode, corresponding to the sloping line in the low frequency region [75,76]. As shown in Figure 8, the EIS patterns of the as-grown film electrodes had no semicircle, and the corresponding impedance was distributed in the range of 2.4–2.9 Ω, which is close to the ohmic impedance of the electrolyte. However, the EIS curves began to show an obvious semicircle, and the semicircle hardly changed with the increase in cycle number. Additionally, the impedance decreased with the gradual increase in O_2_ flow ratios, and the impedance changed from 246 to 130 Ω, which means that the cycled SnO_x_ had a lower charge transfer impedance than pure Sn. The faster dynamic performance is attributed to the fact that the enhanced Li_2_O matrix supplies a highly stable network around the Sn nanoparticles and provides a relatively good electrical contact.

## 4. Conclusions

The SnO_x_ films with various oxygen deficiencies were deposited with different Ar/O_2_ flow ratios using the pDC−RMS method. The main focus was on improving the cycle performance by selecting proper stoichiometry SnO_x_. First, the SnO_x_ films were identified as a mixture of Sn, SnO and SnO_2_ by comparing the Sn 3d and O 1s spectra using the deconvolution procedure. Second, all physical properties (transmittance, crystal structure, density and surface morphology) of the SnO_x_ films showed a sudden transition at the deposition condition of Ar/O_2_ = 26/4, consistent with the turning point for composition evolution. Third, the increase in oxidation degree within the SnO_x_ films determined the reaction transition from “single alloying” to “conversion + alloying” mode. A certain amount of Li_2_O prevented the Li–Sn alloy from suffering mechanical deterioration during repeated discharge/charge cycles, and the SnO_1.71_ film deposited with Ar/O_2_ = 23/7 showed the highest reversible capacity of 396.1 mAh/g at the 50th cycle with a current density of 44 μA/cm^2^.

## Figures and Tables

**Figure 1 materials-14-01803-f001:**
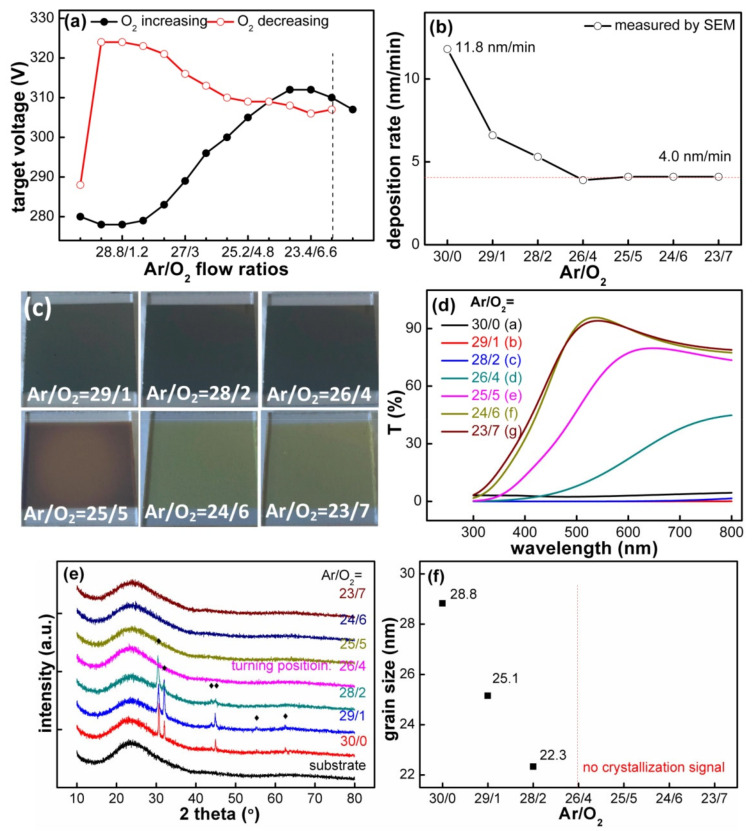
(**a**) Voltage hysteresis curve, (**b**) deposition rate, (**c**) optical images, (**d**) transmittance curves, (**e**) XRD patterns and (**f**) average grain size of the SnO_x_ films deposited with various Ar/O_2_ flow ratios.

**Figure 2 materials-14-01803-f002:**
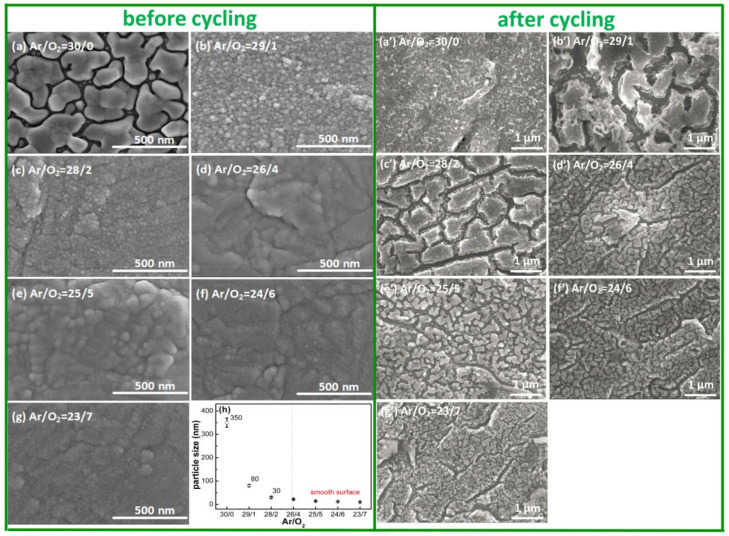
The surface SEM images of the SnO_x_ films before and after cycling, deposited with different Ar/O_2_ flow ratios: (**a**,**a’**) Ar/O_2_ = 30/0, (**b**,**b’**) Ar/O_2_ = 29/1, (**c**,**c’**) Ar/O_2_ = 28/2, (**d**,**d’**) Ar/O_2_ = 26/4, (**e**,**e’**) Ar/O_2_ = 25/5, (**f**,**f’**) Ar/O_2_ = 24/6 and (**g**,**g’**) Ar/O_2_ = 23/7, and (**h**) represents the average particle size on SnO_x_ films.

**Figure 3 materials-14-01803-f003:**
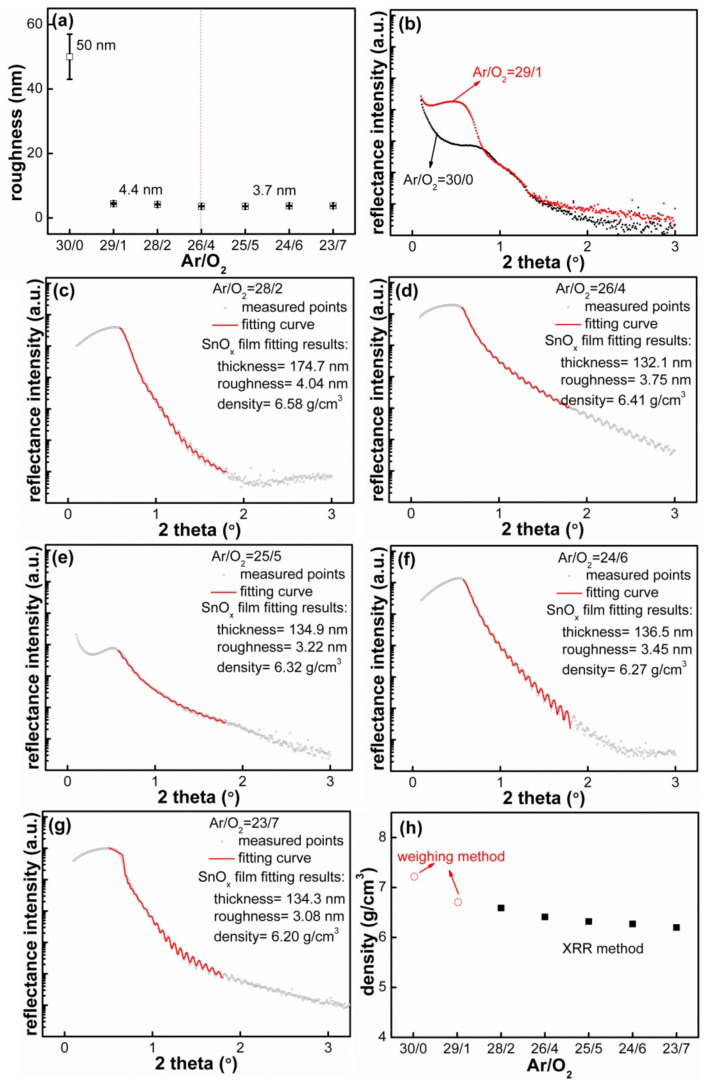
(**a**) Summary of the surface roughness measured by AFM; (**b**) the reflectance curves of the films deposited with Ar/O_2_ = 30/0 and 29/1; the reflectance curves and fitting results of the films deposited with (**c**) Ar/O_2_ = 28/2, (**d**) Ar/O_2_ = 26/4, (**e**) Ar/O_2_ = 25/5, (**f**) Ar/O_2_ = 24/6 and (**g**) Ar/O_2_ = 23/7; (**h**) the film density as a function of Ar/O_2_ flow ratios.

**Figure 4 materials-14-01803-f004:**
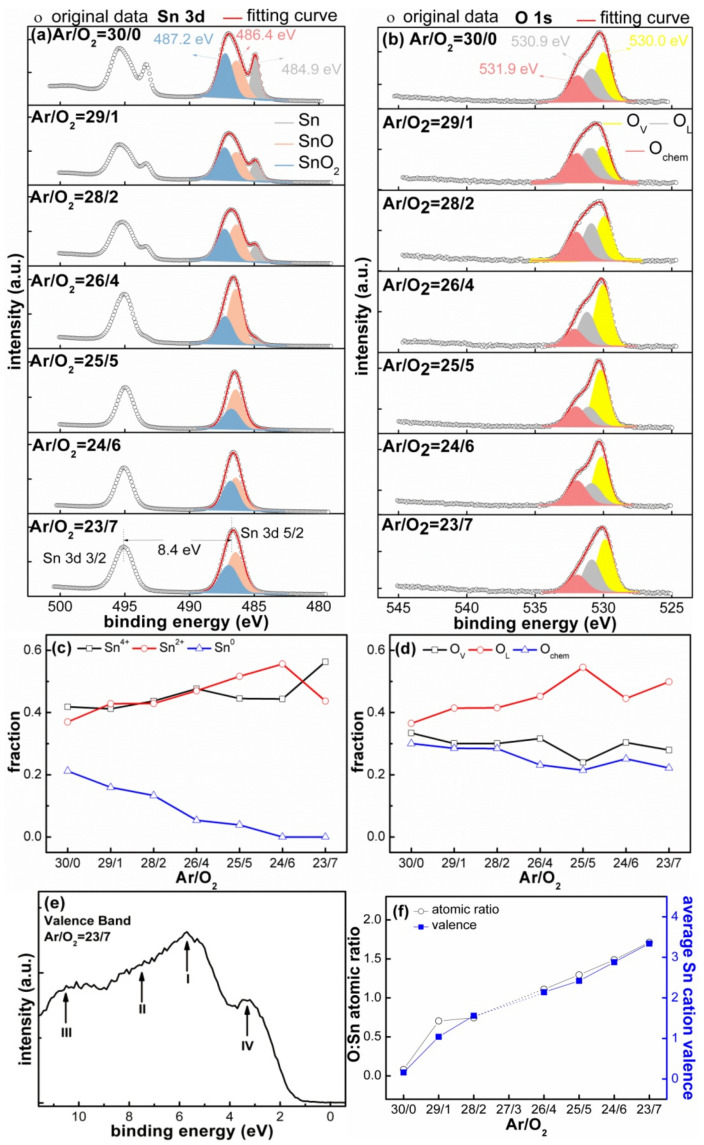
(**a**) Sn 3d and (**b**) O 1s spectra of the SnO_x_ films deposited with different Ar/O_2_ flow ratios; the relative content of (**c**) Sn and (**d**) O with increased O_2_ flow ratios; (**e**) valence band spectrum; (**f**) summary of the O:Sn atomic ratios and average Sn cation valence.

**Figure 5 materials-14-01803-f005:**
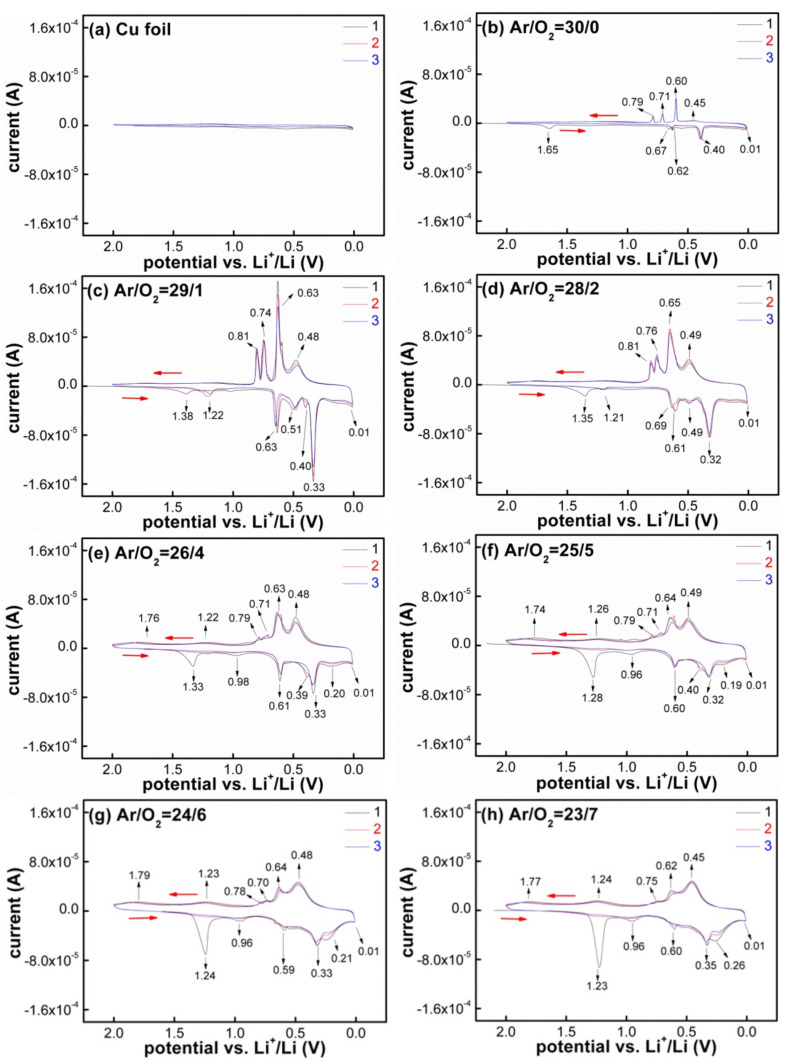
CV curves of the SnO_x_ films deposited with different Ar/O_2_ flow ratios: (**a**) bare Cu foil, (**b**) 30/0, (**c**) 29/1, (**d**) 28/2, (**e**) 26/4, (**f**) 25/5, (**g**) 24/6 and (**h**) 23/7.

**Figure 6 materials-14-01803-f006:**
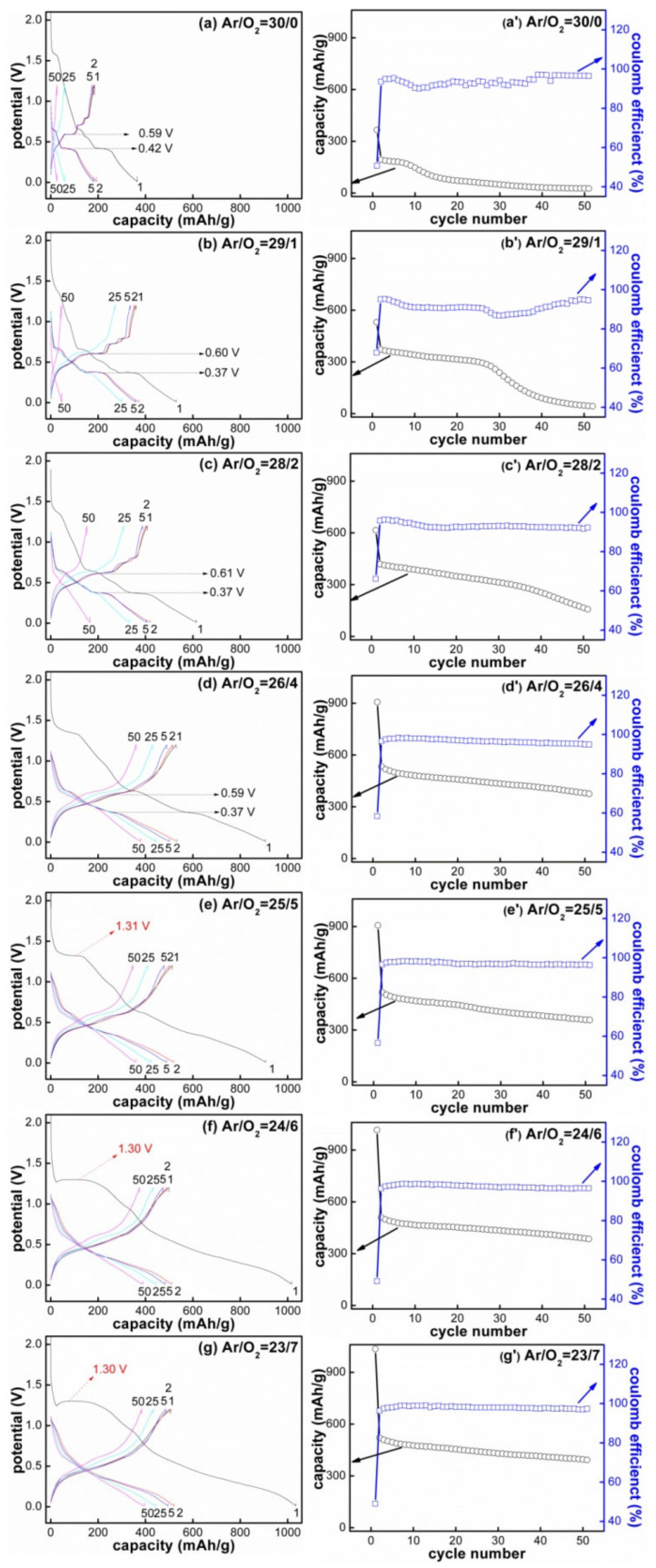
Cycling performance of the SnO_x_ films deposited with different Ar/O_2_ flow ratios: (**a**,**a’**) 30/0, (**b**,**b’**) 29/1, (**c**,**c’**) 28/2, (**d**,**d’**) 26/4, (**e**,**e’**) 25/5, (**f**,**f’**) 24/6 and (**g**,**g’**) 23/7.

**Figure 7 materials-14-01803-f007:**
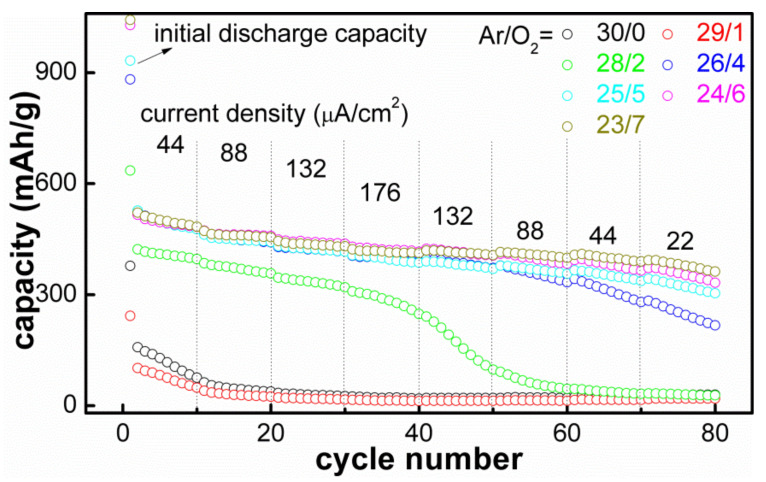
Rate performance of the SnO_x_ films deposited with different Ar/O_2_ flow ratios.

**Figure 8 materials-14-01803-f008:**
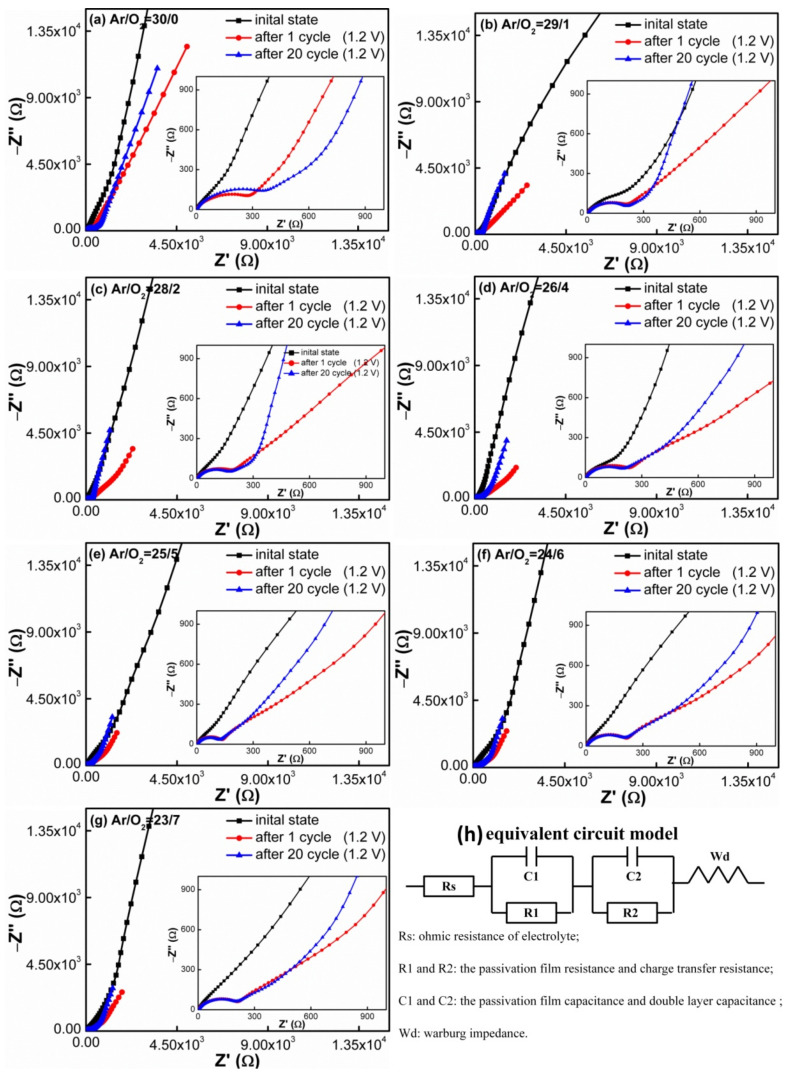
Impedance spectra of the SnO_x_ films deposited at different Ar/O_2_ flow ratios: (**a**) 30/0, (**b**) 29/1, (**c**) 28/2, (**d**) 26/4, (**e**) 25/5, (**f**) 24/6 and (**g**) 23/7. The schematic diagram in (**h**) shows the equivalent circuit.

## Data Availability

Data sharing not applicable.

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
