# Peer review of "Stoichiometry Dependence of Physical and Electrochemical Properties of the SnOx Film Anodes Deposited by Pulse DC Magnetron Sputtering"

_materials, 2021, doi:10.3390/ma14071803_

Round 1
Reviewer 1 Report
A lot of good work has been described in this paper. However, to make it more appealing to readers the authors need to highlight the significance of the work more in the abstract and in the introduction section.
Reviewer 2 Report
Stoichiometry dependence of physical and electrochemical properties of the SnOx film anodes deposited by pulse DC magnetron sputtering
Authors have presented voluminous experimental demonstrations in the structural, optical, morphological, and electrochemical properties of SnOx thin films with the thickness of 110 – 220 nm as an anode used for thin film lithium ion batteries. This manuscript consists of some interesting data with fluent data and discussion; therefore, this reviewer recommends this manuscript to be acceptable for publication in MDPI Materials. However, several comments/questions which need to be addressed before publishing this manuscript as follow:
- 1. Authors need to add a description comparing the pulse DC magnetron sputtering (pDC-MS) method to RF magnetron sputtering (RF-MS) in the introduction section. In addition, it would be helpful for readers if the explanation of reactive sputtering is added.
- 2. The authors sought to control the characteristics of SnOx films by varying the ratio of Ar/O2 flow ratio during the reactive pDC-MS process. Eventually, the authors wanted to examine the effects of stoichiometry in SnOx films caused by it, but was the stoichiometry of SnOx films ‘controllable’ by the change in the ratio of Ar/O2 gas, as represented in Figure 4(f) showing linearity?
- 3. Figures S3(b) and S3(c) show a thicker thin film than those with other conditions. Why did they show unusual results? While the deposition rate in Figure 1(b) appears to have been deposited for 10 min, the deposition time is 34 min if the deposition rate is 4.0 nm/min for the condition of Ar/O2=24/6. Please specify the deposition time by each condition in the experimental section, and also add an explanation of why the authors designed the varied deposition time by condition.
- 4. In relation to Q3, the effects of these different thicknesses will affect the thin film characteristics including the optical properties, which are not clearly described whether thickness-difference has been considered.
- 5. What was the X-ray incident angle for XPS analysis? Isn't it dangerous to conclude the chemical composition of SnOx thin films only with data on the near surface? Is it not necessary to try the XPS depth profile?
- 6. Please write down the terminology and references for formula (1.1). Does ‘ASF’ stand for 'sensitivity factor' as shown in line 327? Figs. 4(c) and 4(d) showed y-axis as fraction, so shouldn't 'relative sensitivity factor (RSF)' be used?
- 7. Why does the transition state appear under Ar/O2=26/4 condition? According to the results of this experiment, this reviewer would like to know the authors' thoughts on whether there is no way to deposition purely Sn metal, even if the metallic Sn target is used for the pDC-MS process with 100% Ar gas flux.
Reviewer 3 Report
The article “Stoichiometry dependence of physical and electrochemical properties of the SnOx film anodes deposited by pulse DC magnetron sputtering” by Y. Ma et al. expose very interesting results on the usage of SnOx nanostructured films as anodes materials for Li battery application. The characterization of these films, synthetized via DC magnetron sputtering, have been carried on using multiple techniques (XRD, XRR, XPS, SEM etc.) demonstrating the existence of an optimal stoichiometry of the anode surface composition in order to maximise the reversible capacity. I recommend this article for publication, after significant changes.
The English of the whole text can be improved and in the following you will find some inaccuracy to modify. I recommend to re-read the text again in search of more errors.
- There is no identification number for the affiliation in relation to the name of any author. Add the number to the names or remove the number from the affiliation
- Line 10: non-stoichiometric SnO2, can make the reader think about slightly O deficient SnO2. Since the stoichiometry of the analysed films vary in a wide range, please replace “non-stoichiometric SnO2” with “Sn oxides”
- Line 14 “et al” in relation to a list of investigation techniques does not sound right, please replace with and more
- Line 14 “involved” does not sound as the appropriate verb to use
- Line 39 “behave” is not correct, did you mean exhibit?
- Line 47 change in “control of film thickness” in “control of the film thickness”
- Line 57-59 Two consecutive phrases begin with hence
- Line 61 “understanding that OF the influence”
- Please specify in the experimental section which kind of curve did you use to fit the XPS spectra and which x-ray source (Al or Mg). Is the valence band spectra of figure 4e acquired with a He lamp or with an x-ray source? Please specify In the text.
- Topography is usually used referring to the study of forms and shapes in 3 dimensions. In this context I think morphology is more appropriate
- Line 164 “regular spherical shape” is not correct. Please replace with sphere-like shape.
- Line 172 “But for the films deposited with low O2 flow ratios, the sputtered particles are quickly oxidized and deposited in the form of Sn-O, hence, they exhibit the characteristics of typical amorphous oxide material” please rephrase this sentence which is not clear
- From line 240 to 245 it is a unique single phrase. Please cut it in smaller phrases to increase the flow of reading and to help to convey the message
- The ASF value for Sn and O, have been calculated? Did you found theme in the literature?
- You calculated the O/Sn=x ratio using equation 1.1. Did you try to calculate the average SnOx valence using the fraction of the fit components (like in doi:10.1016/j.elspec.2004.03.004 equation 1)? This may be a helpful addition for the reader.
- Calculate the stoichiometry of an oxide film is often a complicated task. The complication rises from the fact that often the detector efficiency for different photoelectron kinetic energies in unknown (Handbook of X-ray Photoelectron Spectroscopy: A Reference Book of Standard Data for Use in X-ray Photoelectron Spectroscopy). I think that in order to properly calculate the stoichiometry, the comparison with a reference SnO2 sample is necessary. In this way it is possible to extrapolate this efficiency ratio and use it to calculate the proper stoichiometry of the SnOx films. I think it will be enough to measure the XPS spectrum of SnOx film annealed at high temperature in oxygen atmosphere or air in order to obtain stoichiometric SnO2.
- In multiple papers found in literature, (e.g. : Mater. Chem. A, 2015, 3, 23420: DOI: 10.1039/c5ta07030a, J of Electrochem. Soc. 166 A5308, 2019: DOI: 10.1149/2.0441903jes, Adv. Energy Materials, 2020, doi.org/10.1002/aenm.202000520) the role of the Work function (WF) of the anode material over the performance of the Li-ion device is explored. I cannot find any paper that relate the WF of SnOx films with the device performance and I think this paper will benefit from the exploration of this aspect. In addition, measuring the WF of the investigated films, it is also possible to extrapolate the size of the primary particles using the method described in (Phys. Chem. Chem. Phys., 2020, 22, 6282-6290 DOI: 10.1039/D0CP00216J) and this may be complementary to the XRD identification of crystallite sizes adding important information for the amorphous SnOx.
Round 2
Reviewer 3 Report
The new version of the manuscript has been improved by the authors, increasing the precision and including information necessaries to allow reproducibility. However, before considering this manuscript for publication, additional changes have to be performed and some points have to be clarified.
- Manuscript title and authors’ names are missing in supplementary materials
- Equation 1.1 does not describe a way to calculate O/Sn (x in SnOx) but instead the Sn/O ratio that is equal to 1\x. Please correct this equation.
- The figure 4f has an experimental point missing in correspondence of Ar/O2=27/3. If there are no data available for this gas mixture just connect the points in correspondence of 26/4 and 28/2 with a straight line, like you did for all the other points.
- Despite the work function (WF) is now mentioned in the abstract and main text, a discussion of the role of this parameter and the possibility to tune it via the control of stoichiometry is completely absent. Please insert a paragraph where you elucidate WF dependence on stoichiometry and why it is important to tune this parameter in Li ions device.
- The figure S10 (A5 in your cover letter) and how it is related to the WF of the two SnOx films investigated is obscure to me. It is definitely not a Secondary electrons onset (SEO), since when measured with Al Ka radiation the SEO BE should be found at 1486.6 eV and not 26-27 eV. In addition the background in the BE region 30-34 eV increases, while after a SEO it should be a flat line with zero intensity (e.g. again Chem. Chem. Phys., 2020, 22, 6282-6290 DOI: 10.1039/D0CP00216J, or Coatings 2020, 10, 1026; DOI:10.3390/coatings10111026). In order to properly measure the SEO, it is necessary to negatively bias the sample otherwise it is impossible to measure the photoelectron emitted with 0 eV of kinetic energy. Did you biased your sample? If yes with what voltage? The spectra depicted in Fig. S10 look like the spectrum of Sn 4d (that should be around BE 25)? Is it correct? And how it is supposed to be related to the WF? I think these data should be clarified or removed from the text.
- In the previous cover letter there is a section "New references". Not all the references have been added to the main text or supplementary materials. Is it intentional or those references where just aimed for the cover letter?
